# Hydrodynamic Responses of a Barge-Type Floating Offshore Wind Turbine Integrated with an Aquaculture Cage

Yuting Zhai [1,2], Haisheng Zhao [1,2,*], Xin Li [1,2,*] and Wei Shi [1]

1 State Key Laboratory of Coastal and Offshore Engineering, Dalian University of Technology, Dalian 116024, China; d12006089@mail.dlut.edu.cn (Y.Z.); weishi@dlut.edu.cn (W.S.)
2 School of Hydraulic Engineering, Faculty of Infrastructure Engineering, Dalian University of Technology, Dalian 116024, China
* Correspondence: hzhao@dlut.edu.cn (H.Z.); lixin@dlut.edu.cn (X.L.)

**Abstract:** The dynamic responses of a new structure combining a barge-type floating offshore wind turbine and an aquaculture cage is investigated numerically. First, a 5 MW barge-type floating offshore wind turbine with an aquaculture cage (FOWT-AC) is designed and the numerical model is established in ANSYS-AQWA. The numerical model of the barge-type FOWT-AC is then checked, and the natural periods of the six degrees of freedom motion satisfy the recommendations of the DNV specification. Based on the reasonable model, the comparison study of dynamic responses between the barge-type FOWT-AC and FOWT under the environmental conditions of the South China Sea is carried out, and it is observed that the FOWT-AC produces a basically lower standard deviation of the motion responses. To investigate the new structure of the barge-type FOWT-AC deeply, the analyses of second-order hydrodynamic response, typical environmental conditions and the mooring line breaking scenario are carried out. The simulation results show that the second-order wave loads increase the dynamic response of the barge-type FOWT-AC slightly unless it causes resonance for the structure. In addition, the motion responses of the floating structures increase significantly when the currents are applied, especially when the aquaculture cage is integrated into the barge-type FOWT. When one of the mooring lines connected to the offshore or onshore side of the platform breaks, the presence of the aquaculture cage results in a smaller standard deviation in the motion responses of the coupled structure, which means that the barge-type FOWT-AC structure is more stable.

**Keywords:** offshore wind turbine; aquaculture cage; barge-type; motion response; mooring system

## 1. Introduction

The development and utilization of wind power started to gain attention since the problems of environmental pollution and the energy crisis became serious. The offshore wind turbine is certainly an excellent choice because of the higher speed of the wind and the lower turbulence of offshore wind farms compared to land-based wind farms. The cost of building traditional fixed wind turbines increases rapidly as the wind power development gradually moves to the deep sea [1], thus floating wind turbines are beginning to be studied by scholars in various countries. Compared with the onshore wind turbine, the floating offshore wind turbine (FOWT) is a dynamic system with a higher degree of freedom because it is simultaneously subjected to the wind, currents and wave loads, and is constrained by a mooring system [2].

At present, the main forms of floating offshore wind turbines are the spar-type, semi-submersible type, tension leg type [3], barge-type [4], innovative type platforms [5], etc. The spar-type platform has a large draft and high-water depth requirement, and its advantages are simple structural form and it is unconditionally stable [6]. The semi-submersible type platform has the advantages of simple installation and good towing, while its structure is complicated and expensive to build [7]. The TLP-type platform has been limited in its

development and use due to the high cost of installation [8]. The barge-type platform has the advantages of simple structure, low cost, and long life, particularly suitable for a water depth of 50~100 m due to its low draft feature [9].

As described in the references [10,11], dynamic wave-structure interaction may result in dynamic amplification/reamplification of the motion response of the floating structure; thus, it is important to study the hydrodynamic performance or the dynamic response of these floating platforms under various environmental loads. Utsunomiya et al. [12] developed a dynamic analysis tool to study spar-type offshore wind turbines and compared the analysis results of the software with those of the experiments. The results showed that the developed program has captured the main behavior of the FOWT. The study of Chen et al. [13] found that the semi-submersible FOWT is more sensitive to wave loading compared to Spar-buoy FOWT. However, when vortex excitation motion (VIM) occurs, the simulation underestimates the current load and lateral flow response because the simulation code does not take into account the effects of VIM. Cao et al. [14] found that the pitch resonant responses differed more significantly between the Newman model and the full QTF model compared to the surge resonance response by studying a semi-submersible 10 MW wind turbine. Therefore, the use of the Newman approximation is inaccurate for second-order calculations, and the high-frequency response appears in the tower top shear spectrum only when the full QTF is used. In the past, the hydrodynamic performance of the barge platform had some disadvantages such as poorer stability compared with other platforms; then IDEOL company [15] developed the concept of "damping pool" (called as "moon pool" as well) to improve the hydrodynamic performance, reduce motion response, and lower construction costs of barge-type platforms. Currently, various studies of barge-type wind turbines have begun to grow, and all currently designed barge-type wind turbine platforms are starting to come with a moon pool. For example, a barge-type FOWT with a moon pool had been designed and their numerical results using AQWA agreed well with the experimental results; meanwhile, wave-frequency motion was found to induce motion primarily in the vertical plane [14]. Whether the breakage [16–19] of mooring lines affects the normal operation of the 5 MW barge-type FOWT was explored by analysis tool F2A, indicating that shutdown is a beneficial measure to protect the platform safety from mooring line breakage [20].

The aquaculture farming below the water level is also an effective use of marine resources in addition to the utilization of wind energy above the water. Currently, high-yield deep-sea farming has become a popular industry, and more than 20 countries and regions around the world are dedicated to the field of aquaculture [21]. Norway, as one of the most developed fishing countries in the world, has already been a world leader in the field of aquaculture equipment design with the aquaculture production increasing to 10 times as much as that of 1990. Particularly in recent years, several famous deep-sea farms have accelerated the development of deep-sea aquaculture production. A 79 m diameter rigid semi-submersible fish farm, Arctic Offshore Farming, was proposed in 2017, consisting of a support structure with pontoons and braces, a mooring system and net cages [22]. One of the most famous aquaculture farm is the OceanFarm1 [23] developed by SalMar in Norway.

The fish farms are subject to currents and waves causing motion responses or fatigue damage to the aquaculture cage; thus, a hydrodynamic analysis of the aquaculture cage is essential. By comparing the numerical results and experimental results in tanks, a method for the structural analysis of aquaculture nets was developed and it was concluded that the resistance load and cage volume depended on the size and weight of the net system [24]. Based on the lumped-mass method and rigid-body kinematics theory, Xu et al. [25] established a mathematical model of a gravity cage system attacked by irregular waves and simulated the hydrodynamic response of the cage system. It was found that the cage system had no significant tumbling motion, and the cage system tended to self-profile against longer waves at high frequencies.

Currently, the upfront investment in wind power is high, and the annual income from power generation alone is still far less than the total investment in offshore wind farms. In order to compensate it, some scholars have come up with the idea of combining offshore wind turbines with aquaculture cage structures. For example, He [26] proposed the multi-use platform concept of combining 10 MW wind turbines with aquaculture in 2015. The earlier research on floating offshore wind turbines with a fish farming cage in China was conducted at Tsinghua University [27]. In order to explore the motion performance, a floating offshore wind turbine with a steel fish-farming cage (FOWT-SFFC) was designed, and the random response and nonlinear dynamic performance of the structure were studied simultaneously; it is noted that the large motion of the pitch has little effect on the nonlinearity of the viscous drag and the mooring interaction [27,28]. Chu and Wang [28] were concerned with the hydrodynamic response of a novel offshore fish farm in the combined floating spar wind turbine and fish cage. An offshore floating multi-purpose platform supporting a 10 MW wind turbine on one side of this barge platform with a fish cage was designed for Blue Growth Farm. Numerical simulation results showed that the dynamic responses of the multi-purpose platform were excellent [29,30]. Moreover, there are other multi-purpose platform concepts that combine wind turbines, aquaculture systems and wave energy converters [31,32] or solar energy [33] into these multi-purpose structures to synthesize various resources.

The coupled structure of a wind turbine with an aquaculture cage can not only use the sea space efficiently, but can also achieve greater benefits with a smaller investment. However, this new type of coupling structure is limited in its development because of its novel orientation. Therefore, it is necessary to propose a reasonable new structure of floating offshore wind turbine integrated with an aquaculture cage (FOWT-AC). Currently, most of the studies on this coupled structure are based on semi-submersible or spar type platforms, whereas the barge-type platform is convenient for transportation, simple in structure, and large in platform area, which is also conducive to the installation of aquaculture control unit supplying constant power for remote and autonomous fish farming operations. There is little research on the barge-type floating offshore wind turbines integrated with an aquaculture cage. Therefore, a novel coupled structure combining a barge-type FOWT and an aquaculture cage is designed in this study.

Numerical simulation is an effective means in analyzing the hydrodynamic characteristics of floating structures, and various numerical methods have been developed for the wave-structure interaction of flexible, floating or moving marine structures. For example, the traditional Boundary Element Method (BEM) [34] can be used to calculate the linear steady-state response of large floating bodies in regular waves; the Smoothed Particle Hydrodynamics (SPH) method [35] can solve the distortion and stretching of the medium by solving partial differential equations, and even combine it with the Finite Element Method (FEM) or BEM to evolve into the coupled SPH-FEM/DEM [36,37]; furthermore, the Finite Volume Method (FVM) [38], Finite Difference Method or even arbitrary Lagrangian-Eulerian Method [39] are widely used for the flow field analysis or dynamic analysis of marine structures. To investigate the hydrodynamic responses of the integrated structure and the effect of the presence of the aquaculture cage on the structure, the FEM-based numerical model is developed by ANSYS-AQWA and the numerically simulated results for the barge-type FOWT-AC and barge-type FOWT are compared in detail.

The remainders of this paper are organized as follows: Section 2 introduces the theoretical methods used in this research; Section 3 presents descriptions of the FOWT-AC, simulation environment and model validation. In Section 4, the results and discussion of this study are reported. Section 5 presents the conclusions of this work. Finally, Section 6 presents the future work.

## 2. Theoretical Methods

The FOWT-AC is a complex multi-body coupled structure with six freedom degrees, which is mainly subject to the environment loads of wind, waves and currents. The

interaction of the coupled structure and environmental loads results in a complicated dynamic response, including elastic deformation of the blades and tower, the rigid body motion of the platform, the dynamic response of the mooring system, and so on. Therefore, the analysis of the coupled structure involves hydrodynamics, structural dynamics, the control of the wind turbine and many other aspects of the theoretical methods.

### 2.1. Equation of Motion in the Time Domain

To calculate the responses of such a complex system, which is regarded as a rigid body in this paper, the time-domain equation of the FOWT-AC can be obtained in the following equation when the three-dimensional potential flow theory is used,

$$(M + A_\infty)\ddot{x}(t) + \int_0^t K(t - \tau)\dot{x}(\tau)d\tau + Cx(t) = F_{wind}(t) + F_{wave}(t) + F_{curr}(t) + F_{moor}(t) \tag{1}$$

where $x$, $\dot{x}$ and $\ddot{x}$ represent the six degrees of freedom (DOFs) displacement, velocity and acceleration of the FOWT-AC platform; $M$ is mass matrix; $A_\infty$ is the infinite-frequency added mass matrix; $K$ represents the wave-radiation-retardation kernel matrix; $C$ is defined as hydrostatic stiffness matrix; $F_{wind}$ is the wind loads on the FOWT-AC obtained by wind speed-thrust curve; $F_{wave}$ and $F_{cuur}$ are the wave excitation load and current force, respectively, which are calculated by potential flow theory and Morison equation; $F_{moor}$ represents the restoring force of the mooring system.

$$K(t) = \frac{2}{\pi} \int_0^\infty b(\omega) \cos(\omega t)d\omega \tag{2}$$

where $b$ is the linear radiation damping matrix.

### 2.2. Potential Flow Theory

The total excitation loads which come from incident waves act on the FOWT-AC structure including first- and second-order wave excitation forces. If the wave steepness of the incident wave is small relative to the size of the structure such as the platform of FOWT-AC, the velocity potential can be written in the form of a series expansion:

$$\Phi(x, y, z, t) = \Phi^{(1)}(x, y, z, t) + \Phi^{(2)}(x, y, z, t) + \cdots + \Phi^{(n)}(x, y, z, t) + \cdots, \tag{3}$$

where $\Phi^{(1)}$, $\Phi^{(2)}$, $\Phi^{(n)}$ are the first-order, second-order and $n$th-order velocity potentials, respectively.

For any sea state, it is assumed that the waves are superimposed by Airy linear waves. The first-order total velocity potential can also be seen as a superposition of the first-order velocity potentials of different frequency wave components, the specific formula of which is expressed as follows:

$$\Phi^{(1)}(x, y, z, t) = \text{Re}\sum_j \phi_j^{(1)}(x, y, z)e^{i\omega_{wave,j}t}, \tag{4}$$

where $\omega_{wave}$ is the wave frequency; $i$ and $j$ are the $i$th and $j$th incident wave.

Similarly, the expression for the second-order total potential function containing sum and difference frequencies is as follows:

$$\Phi^{(2)}(x, y, z, t) = \text{Re}\sum_j \sum_i [\phi_{ji}^+(x)e^{i(\omega_{wave,j}+\omega_{wave,i})t} + \phi_{ji}^-(x)e^{i(\omega_{wave,j}-\omega_{wave,i})t}], \tag{5}$$

Once the potential function is obtained, the hydro-dynamic pressure ($p$) at any point of the flow field can be obtained according to Bernoulli's Equation (6), and the wave force acting on the structure can be obtained by integration:

$$\frac{\partial \Phi}{\partial t} + \frac{1}{2}(\nabla \Phi) \cdot (\nabla \Phi) + gz + \frac{p}{\rho} = 0, \tag{6}$$

$$\boldsymbol{F}_{wave} = \int_{S_B} p(x, y, z, t) \boldsymbol{n} dS, \tag{7}$$

where $S_B$ is the wet surface of the structure at any moment; $p$ is the hydro-dynamic pressure; $\boldsymbol{n}$ is the normal vector of the structure surface.

### 2.3. Mesh Grouping Method

Due to the large size of the aquaculture cage, the large number of netting lines increase the modeling difficulty and also grow the computational volume significantly when the nets are modeled according to the prototype. Therefore, most of the netting models use the mesh grouping method to reduce the number of netting lines. In order to ensure the accuracy of the calculations after mesh grouping, the following principles need to be observed:

(1) The nets should be kept equal in quantity before and after grouping.
(2) Similar geometric scales should be maintained before and after grouping.
(3) The nets before and after grouping should be kept equal to the hydrodynamic force.

The above three conditions can be expressed as three equations as follows:

$$M_{grouping} = M, \tag{8}$$

$$A_{grouping} = \sum A, \tag{9}$$

$$(EA_{\sec tion})_{grouping} = \sum EA_{\sec tion}, \tag{10}$$

where $M$ for the quality of the netting; $A$ and $A_{\sec tion}$ represent the projected area and cross-sectional area of the netting; and $E$ represents the Young's modulus of the netting.

The paramount characteristic for netting without knots is the solidity ratio ($S_n$). The $S_n$ is defined as the area of nets to the area of the plane. From Figure 1, the solidity ratio can be calculated as follows:

$$S_n = \frac{2d}{\lambda} - \left(\frac{d}{\lambda}\right)^2, \tag{11}$$

where $\lambda$ represents the mesh size, and $d$ is rope diameter.

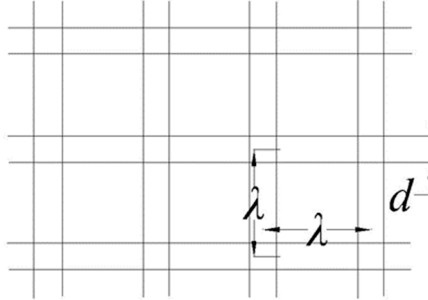

**Figure 1.** Equivalent netting.

In order to ensure that the hydrodynamic forces before and after grouping are similar, it is necessary to ensure that the real nets and the equivalent nets have the same $C_d$ and densities $S_n$.

### 2.4. Hydrodynamic Force of Netting

The two main simplified models for calculating the hydrodynamic force of netting are the Morison model and the Screen model. The Screen model is originally developed for steady flow, while the Morison model is for a wider range of situations. As opposed to potential flow theory, the Morison equation is the method that applies when the diameter of the structure is small relative to the wavelength. Therefore, the Morison equation is adopted herein for simulating the hydrodynamic force of the netting and columns of the cage due to their diameter being small relative to the wavelength. The Morison equation can be written as the following expression,

$$F(t) = \frac{1}{2}\rho C_d D u(t)|u(t)| + (1 + C_m)\rho\frac{\pi D^2}{4}\dot{u}(t), \tag{12}$$

where $u(t)$ is the horizontal velocity at the wave water particles; $\dot{u}(t)$ is the horizontal acceleration at the wave water particles; $C_m$ is the additional mass coefficient, and $C_d$ is the drag coefficient.

Because the quality of the netting is quite small, the inertial forces on the netting are small, and only drag forces are considered [40]. A Morison model force diagram for two orthogonal equivalent net lines is shown in Figure 2, where $A$ is the projected area of the net line; $U_{rel}$ is the horizontal relative velocity of water particles and structure. By simply superimposing the drag force vectors on all the net lines, the drag force applied to the entire netting can be obtained.

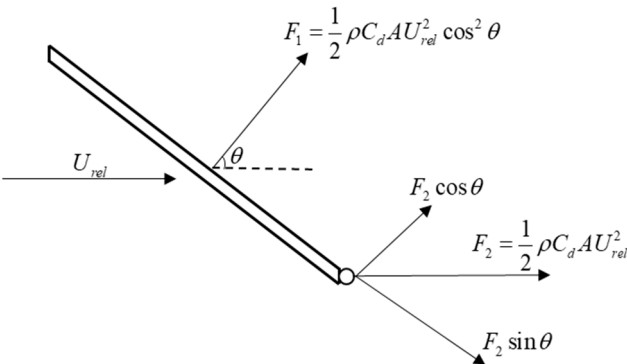

**Figure 2.** Force diagram for the Morrison model of orthogonal lines.

### 2.5. Mooring Lines Tension

Because the quasi-static method considers the steady and slow drifting motion of the floating structure by ignoring the dynamic effects, the quasi-static calculation can shorten calculation period and guarantee a certain accuracy when inertia forces have less influence. In the numerical simulation of floating wind turbines, the quasi-static method is already a mature method. For example, Zhao et al. [41,42] used this method in a semi-submersible wind turbine, and Yu et al. [43] employed it in the spar wind turbine, and reasonable numerical results were obtained. Hence, the mooring lines tension is calculated based on the quasi-static equilibrium method in this study, and the mass of the line, elastic tension and subsea friction are considered while the bending stiffness, the inertia and the damping of the mooring system are ignored [44]. It is beneficial for the safe design of the integrated structure since relatively conservative results are predicted.

A single mooring line with undercover length in a local coordinate system are shown in Figure 3. The anchor is the origin and the positive direction of the z-axis is vertical. The horizontal and vertical components of mooring lines effective tension $(H_F, V_F)$ at the fairlead are calculated by the following equation [43]:

$$X_F(H_F, V_F) =$$
$$L - \frac{V_F}{\omega} + \frac{H_F}{\omega} \ln\left[\frac{V_F}{H_F} + \sqrt{1 + (\frac{V_F}{H_F})^2}\right] + \frac{H_F L}{EA_m} + \frac{C_B \omega}{2EA_m}\left[-(L - \frac{V_F}{\omega})^2 + (L - \frac{V_F}{\omega} - \frac{H_F}{C_B \omega})\cdot\max(L - \frac{V_F}{\omega} - \frac{H_F}{C_B \omega}, 0)\right] \quad (13)$$

$$Z_F(H_F, V_F) = \frac{H_F}{\omega}\left[\sqrt{1 + (\frac{V_F}{H_F})^2} - \sqrt{1 + (\frac{V_F - \omega L}{H_F})^2}\right] + \frac{1}{EA_m}(V_F L - \frac{\omega L^2}{2}), \quad (14)$$

$$F_{moor} = \sum \sqrt{H_F^2 + V_F^2}, \quad (15)$$

where $L$ is the length of the mooring line; $\omega$ is the density of the mooring line in the water; $A_m$ is the cross-sectional area of the mooring line; $EA_m$ is the extensional stiffness; $C_B$ is the friction coefficient of the seabed connection section of the mooring line.

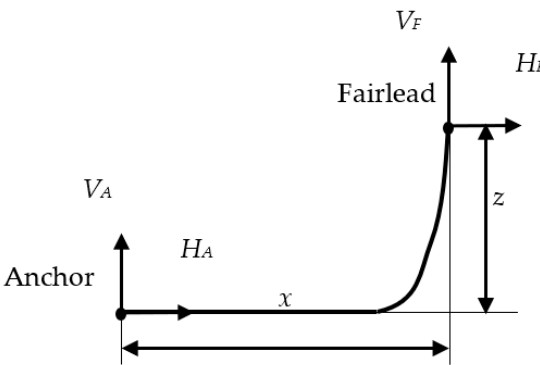

**Figure 3.** Mooring system model.

## 3. Numerical Model and Model Testing

### 3.1. Design of the FOWT-AC

The overall conceptual diagram of the barge-type FOWT-AC for this study is shown in Figure 4. The barge-type platform adopted refers to a similar concept in reference [16], which supports the National Renewable Energy Laboratory (NREL) 5 MW baseline wind turbine. Their specific parameters are shown in Tables 1 and 2. The aquaculture cage is directly connected to the bottom of the barge platform, which comprises eight side columns, sixteen diagonal columns and nets, forming a square shape cage with internal space of 60 m length and 30 m height, as shown in Figure 5. The wind turbine and the aquaculture cage share the platform to build workplaces, so the aquaculture cage can use the buoyancy of the platform as well as the electricity generated by the wind turbine directly for operation and maintenance. The selection of nets refers to the parameters of Econet of OceanFarm1 [23]. The solidity ratio $S_n$, mesh size $\lambda$ and rope diameter $d$ of the aquaculture cage are set to 0.16, 5 m and 0.4 m. The range of the damping coefficient in the model of the aquaculture cage is [0.4–1.2] [23], which is mainly determined from the Reynolds value. The value used in this study is 0.8 [23] according to the common sea state. The main parameters of the aquaculture cage can be found in Table 3. The designed water depth of the numerical simulation is selected as 50 m. In this case the mooring system belongs to a shallow water mooring. Due to the complexity of the shallow water mooring, eight mooring lines are applied to keep the coupled structure in position, which features two sets of upwind lines and two sets of downwind lines. The mooring line adopts a gearless catenary type, and it is Grade 4. Table 4 lists the detailed parameters of the mooring system.

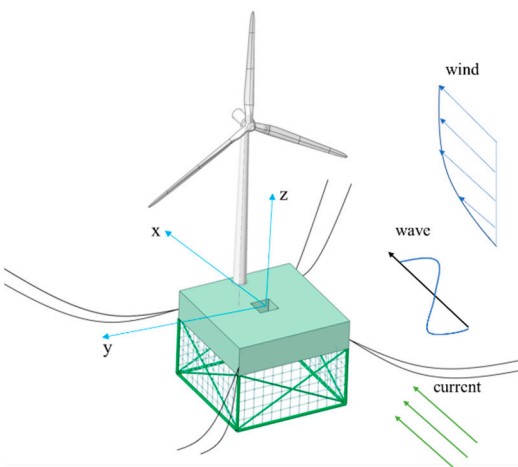

**Figure 4.** Conceptual diagram of the barge-type FOWT with aquaculture cage.

**Table 1.** Main parameters of the barge-type platform.

| Parameters | Value |
|---|---|
| Barge size (W × L × H) (m) | $60 \times 60 \times 15$ |
| Moonpool (W × L × H) (m) | $10 \times 10 \times 15$ |
| Draft (m) | 10 |
| Mass (kg) | 32,200,282 |
| COG (m) | $(-0.37, 0, -4.86)$ |
| Roll inertia (kg/m$^3$) | 9,660,000,000 |
| Pitch inertia (kg/m$^3$) | 9,660,000,000 |
| Yaw inertia (kg/m$^3$) | 19,300,000,000 |

**Table 2.** Specifications of the NREL 5 MW wind turbine.

| Parameters | Value |
|---|---|
| Rotor, nacelle, tower mass (t) | 110, 240, 347.46 |
| Hub height (m) | 90 |
| COG (m) | (20.0, 0.0, 70.6) |
| Rotor, hub diameter (m) | 126,3 |
| Roll inertia (kg/m$^3$) | 240,000,000 |
| Pitch inertia (kg/m$^3$) | 240,000,000 |
| Yaw inertia (kg/m$^3$) | 480,000,000 |
| Rotor, nacelle, tower mass (t) | 110, 240, 347.46 |

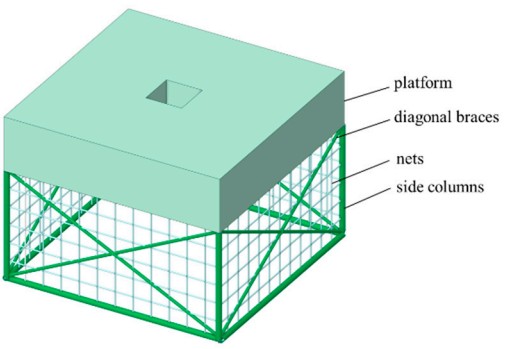

**Figure 5.** Details of the aquaculture cage.

**Table 3.** Main parameters of the aquaculture cage.

| Parameters | Value |
|---|---|
| Diameter of side column, diagonal column and equivalent net (m) | 1.5, 0.8, 0.4 |
| The thickness of the side column and diagonal column (m) | 0.02 |
| Diameter of the equivalent net (m) | 0.4 |
| Distance of adjacent equivalent net (m) | 5 |
| Weight of equivalent net (g/m2) | 590 |

**Table 4.** Properties of the mooring system.

| Parameters | Value |
|---|---|
| Number of mooring lines | 4 × 2 |
| Anchor between adjacent lines | 5°/85° |
| Water depth (m) | 50 |
| Radius to anchor form the platform centerline (m) | 408 |
| Unstretched mooring line length (m) | 375 |
| Mooring line diameter (m) | 0.162 |
| Catalog breaking load (kN) | 9319 |
| Equivalent mooring line mass density (kg/m) | 522.25 |

### 3.2. Numerical Model

These coupled time-domain analyses for the barge-type FOWT-AC can be conducted by ANSYS-AQWA. ANSYS-AQWA [45] is mainly based on the Morison equation and 3D potential flow theory for simulation analysis, which allows the user to apply histories via dynamic link libraries and calculate the motion position, velocity and force of the structure accordingly. However, the 3D effects generated by the diagonal columns in the aquaculture cage are not captured by the simulation. Besides, the effect of wave energy loss on the onshore columns of the aquaculture cage is ignored.

Since the focus of this study is on the hydrodynamic performance and motion response of the platform, the wind turbine only plays the roles of gravity and the transfer of wind thrust and moment from the blades to the platform, so the upper parts of the structure can be simplified. In this work, the blades are simplified to a disk with the same wind sweeping area, and the dynamic link library is used to apply the thrust force and wind tilting moment that vary in real-time with the relative wind speed. The speed-thrust curve is shown in Figure 6. And the mesh model of the numerical simulation simplified is presented in Figure 7.

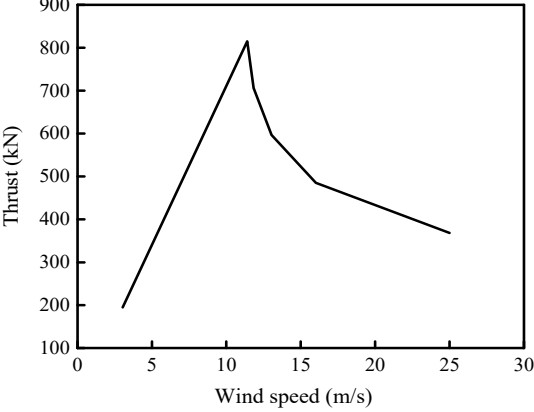

**Figure 6.** Diagram of the wind speed-thrust curve.

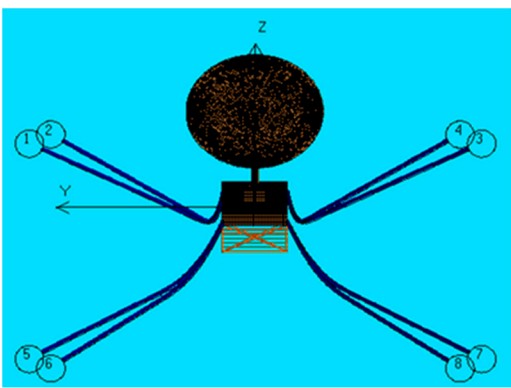

**Figure 7.** The mesh model of the numerical simulation simplified, where mooring line 1 to 8 are marked.

The flow chart of numerical simulation for barge-type FOWT-AC is illustrated in Figure 8. The FOWT-AC structure is composed of four parts from top to bottom, namely, wind turbine, barge platform, aquaculture cage and mooring system, and is coupled by three external loads of wind, wave and currents with its own structural motions. The wind turbine and above-water parts of the platform are subjected to the aerodynamics, while the underwater parts of the platform, aquaculture cage and mooring system suffer hydrodynamics. As indicated previously, the wind loads are set by the dynamic link library using user-force; the hydrodynamics of the platform and aquaculture cage are simulated by the potential flow theory and Morison equation; the tension of the mooring system is calculated by quasi-static theory [45]. Moreover, the operating state will be determined by the control system according to the dynamic response state of the wind turbine. The coupling effects mentioned in the above determine the final dynamic characteristics of the integrated FOWT-AC system.

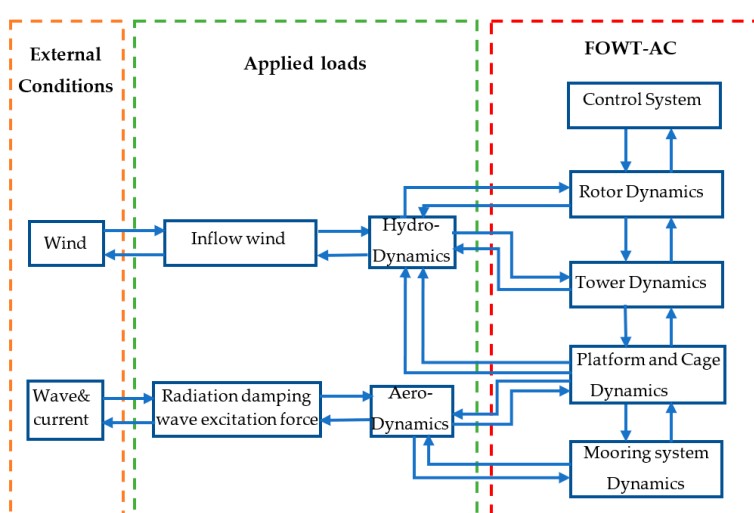

**Figure 8.** Flow chart of numerical model.

*3.3. Environmental Conditions*

To investigate the effect of hydrodynamic performance of the barge-type FOWT-AC, the environmental conditions (ECs) in this study are selected mainly by using the irregular wave, steady wind and current. And they are divided into eight ECs, as detailed in Table 5. The JONSWAP spectrum is used to generate the irregular wave time series by indicating the significant wave height ($H_s$) and spectral peak period ($T_p$). Three values of wave height are selected including the normal and extreme sea states in the South China Sea. And three values of wind speed are selected: below rated wind speed, rated wind speed and greater

than rated wind speed. Two current conditions with and without currents are set up to investigate the effect of currents on the structure. Each simulation lasts for 4200 s, and the results of the first 600 s are eliminated to mitigate the startup transient effects. The wind, current and wave directions are input along the positive x-axis in the numerical calculations just as Figure 4. The schematic layout of mooring lines is illustrated in Figure 9. The ratio of wavelength-to-structural length (or width) has a critical effect on the hydrodynamic loads applied on a marine platform [39]; for example, the Morison equation is generally used when the ratio of the characteristic size of the structure to wavelength is less than 0.2, while the potential flow theory should be used for large size structures when the effect of radiation damping cannot be neglected. In this shallow water case, according to the dispersion relationship, it is known that the wavelength range in the above wave period cases is between 115 m and 210 m. The ratios of characteristic length of the columns of cage to the wavelength are between 0.004 and 0.013; thus, the Morison equation method is employed for the columns of the cage, while the potential flow theory is adopted for the hydrodynamic analysis of the barge-type platform due to the ratios of structural length to wavelength being located between 0.3 and 0.52.

**Table 5.** Environmental conditions.

|  | Wind Speed (m/s) | $H_s$ (m) | $T_p$ (m) | Current Speed (m/s) | Number of the Broken Line | Turbine Status |
|---|---|---|---|---|---|---|
| EC1 | 11.4 | 3.2 | 5.7 | 1 | no | Operating |
| EC2 | 8 | 3.2 | 5.7 | 1 | no | Operating |
| EC3 | 25 | 3.2 | 5.7 | 1 | no | Shutdown |
| EC4 | 11.4 | 1.67 | 5.17 | 1 | no | Operating |
| EC5 | 11.4 | 5.52 | 9.4 | 1 | no | Operating |
| EC6 | 11.4 | 3.2 | 5.7 | 0 | no | Operating |
| EC7 | 11.4 | 3.2 | 5.7 | 1 | #3 | Operating |
| EC8 | 11.4 | 3.2 | 5.7 | 1 | #8 | Operating |

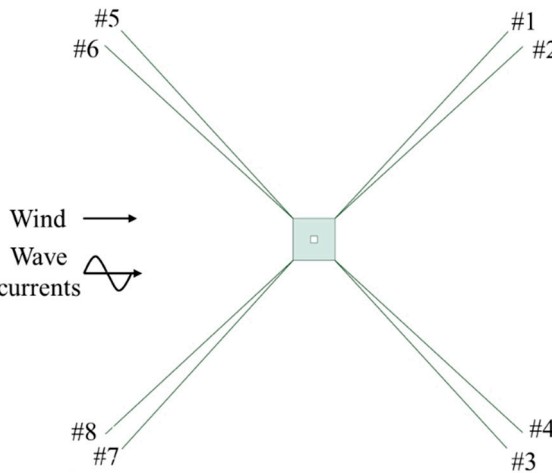

**Figure 9.** The schematic layout of mooring line 1 to 8.

### 3.4. Model Validation

Because of the powerful performance of ANSYS-AQWA, some researchers have chosen it to study FOWT. These models were built with a panel model of the large structure, Morison elements of the slender structure, and a quasi-static theory of the mooring system. Nallayarasu and Saravanapriya [46] built a Spar-type wind turbine model by ANSYS-AQWA and compared the numerical simulation results of the model with the experimental results and concluded that the experimentally measured RAOs agrees very well with the numerically simulated results with a maximum difference of 15%. By comparing the results of the 1:30 experimental model with the numerical model built using ANSYS-AQWA, Ruzzo



et al. [47] noted that the experimental results in the peak frequency of the RAO were similar to the numerical prediction. Rajeswari and Nallayarasu [48] compared hydrodynamic responses of the experimental and the numerical results of the semi-submerged floaters with three columns and four columns, and a difference of less than 10% except the surge motion was observed. Based on the studies of these scholars it can be determined that the numerical modeling of floating wind turbines can get excellent results through the modeling theory of ANSYS-AQWA. For the new structures of floating wind turbine with a fish cage, Chu and Wang [28] modeled the Spar-type platform with a fish cage by means of ANSYS-AQWA and obtained reasonable results. The above references verify the validity of the theoretical methods implemented in ANSYS-AQWA in performing the hydrodynamic analyses of the floating wind turbine and the integrated structure.

To guarantee the mesh quality of the numerical model for the barge-type FOWT-AC, the comparison of the second-order drift force between the near-field and far-field is carried out. This is because the near-field method employs the potential flow theory to resolve the second-order drift force by integrating over the surface of a wet element, whereas the momentum theorem is adopted by the far-field method. Therefore, the near-field results are dependent on the mesh quality. When the results of the near-field and far-field methods follow the same trend with little difference, the mesh for hydrodynamic calculations of the floating structure is considered to meet the requirements. Figure 10 shows the comparison of second-order drift force results for the barge-type FOWT-AC and barge-type FOWT in the near and far fields. It can be seen that the results for the near and far fields of the same structure are approximately identical, representing the reliability of the mesh quality.

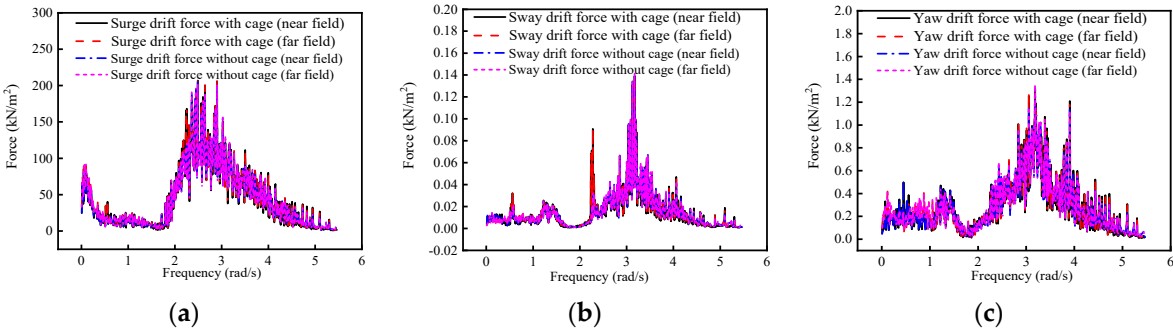

**Figure 10.** Comparison of second-order drift force results for the barge-type FOWT-AC and FOWT in the near and far fields: (**a**) Comparison of second-order surge drift force; (**b**) Comparison of second-order sway drift force; (**c**) Comparison of second-order yaw drift force.

After the numerical model is affirmed, AQWA-LINE is used to calculate the natural periods of heave and pitch, and the free decays of the surge and yaw are conducted by AQWA-DRIFT to determine their natural periods for the floating structures with and without the aquaculture cage, respectively. As can be seen from the results presented in Table 6, the natural periods of several degrees of freedom have increased due to the presence of the aquaculture cage, except for the yaw. The change of these periods is mainly due to the increase in the overall weight of the structure and the downward shift of the center of gravity. In addition, the natural periods of these two integrated structures are in accordance with the range of natural periods of barge-type FOWT platforms proposed by the DNV-RP−0286 [49] specification, indicating that the design of the floating structures is reasonable.

**Table 6.** The natural periods of the barge-type FOWT-AC and barge-type FOWT.

|  | With Cage | Without Cage |
|---|---|---|
| Surge | 45.59 s | 44.28 s |
| Heave | 9.98 s | 9.96 s |
| Pitch | 10.82 s | 10.09 s |
| Yaw | 51.66 s | 51.66 s |
| Surge | 45.59 s | 44.28 s |

## 4. Results and Discussion

### 4.1. Effect of Second-Order Wave Loads on Motion Responses of Coupled Structures

Second-order wave loads are irregular forces composed of three components, such as second-order mean drift force, sum- and difference-frequency wave loads. Although second-order wave loads are smaller in magnitude than first-order wave loads, they can cause large drifts in marine structures if applied over a long period of time. And the sum- and difference-frequency wave loads can excite the floating structures' eigen-frequencies, resulting in significant oscillation, which may cause damage to the floating structures. Therefore, the study of second-order wave-excitation forces transfer functions (QTFs) is quite necessary.

Second-order difference-frequency and sum-frequency QTFs of the barge-type FOWT with and without an aquaculture cage for a zero-degree wave heading are presented in Figures 11 and 12. 100 frequencies within 0~2 rad/s are considered in the calculation of wave-excitation forces transfer functions, thus the dimension of the QTF is $100 \times 100$. It is noted that the QTFs of the barge-type FOWT are more evenly distributed than those of the barge-type FOWT-AC with smaller maximum magnitude values except for difference-frequency QTFs of heave. Although the maximum value of the sum-frequency is greater than the difference-frequency, the overall QTFs of the sum-frequency is distributed in a relatively small amplitude range. Therefore, the sum-frequency has less influence on these coupled structures than the difference-frequency. Besides, the full-field difference-frequency QTFs graph exhibits an apparent symmetry about the anti-diagonals (solid black line), which is related to the mean-drift load at $\omega_i = \omega_j$, thereby resulting in a static load.

Figure 13 shows a comparison of motions for the barge-type FOWT-AC and barge-type FOWT at first order and first order with QTF conditions to investigate the effect of second-order wave forces on floating structures. It is found in Figure 13a that the presence of the second-order wave loads makes the mean value and standard deviations of the surge smaller. Figure 13b shows that the presence of the second-order wave loads makes the mean value of the pitch larger and the standard deviations of the pitch smaller. From Table 7, it can be seen that the changes caused by second-order wave forces are less than 3% both in mean and standard deviation of pitch, while the changes of standard deviation of the surge are up to about 20%. The variation of the motions is due to the Morison drag force coming from the aquaculture cage. But the effect of second-order wave loading on the responses of the coupled structures is not on the same order of magnitude as the first-order wave loading. Therefore, the accuracy of the results of first-order wave forces is sufficient in the subsequent analysis.

**Table 7.** Percentage comparison results of motion response under 1st and 1st +QTF wave loads.

|  | Mean of Surge | Std of Surge | Mean of Pitch | Std of Pitch |
|---|---|---|---|---|
| With cage | 2.88% | 19.88% | 2.73% | 0.77% |
| Without cage | 3.51% | 21.13% | 1.95% | 2.00% |

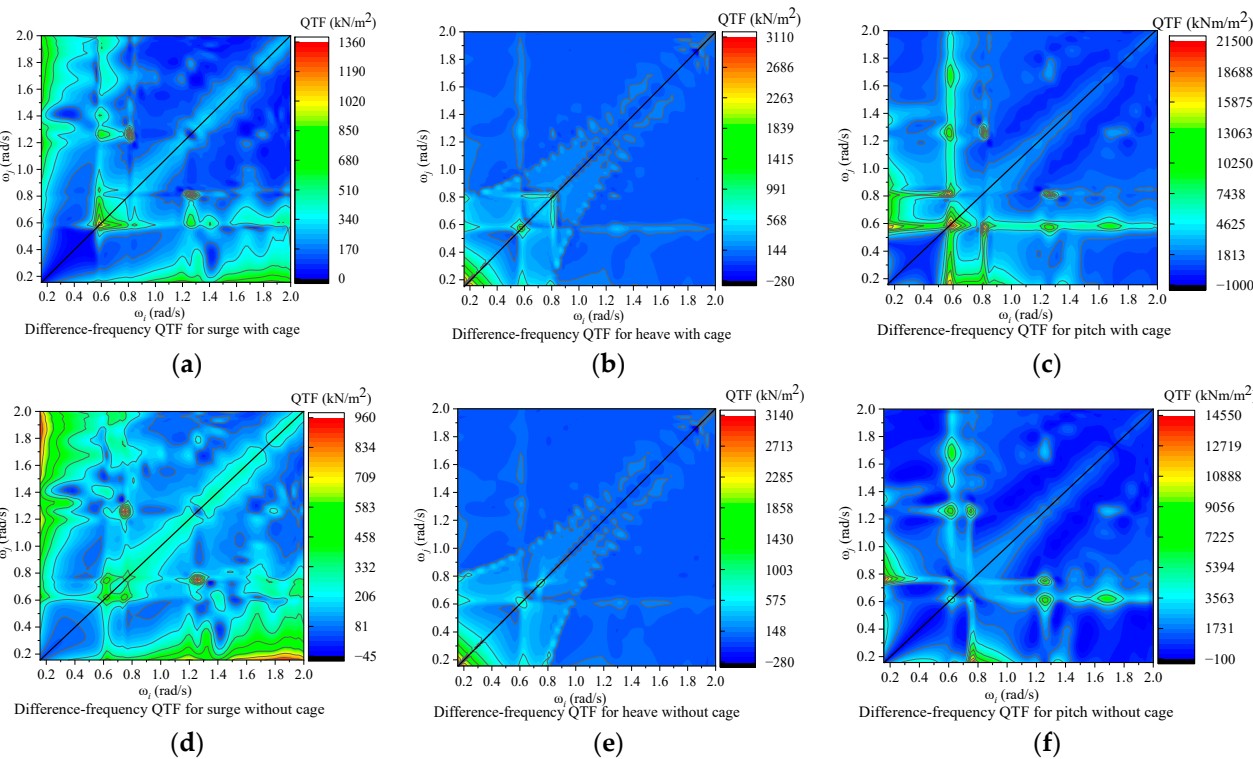

**Figure 11.** Second-order difference-frequency QTFs of the barge-type FOWT-AC and barge-type FOWT for a zero-degree wave heading: (**a**) Surge of FOWT-AC; (**b**) Heave of FOWT-AC; (**c**) Pitch of FOWT-AC; (**d**) Surge of FOWT; (**e**) Heave of FOWT; (**f**) Pitch of FOWT.

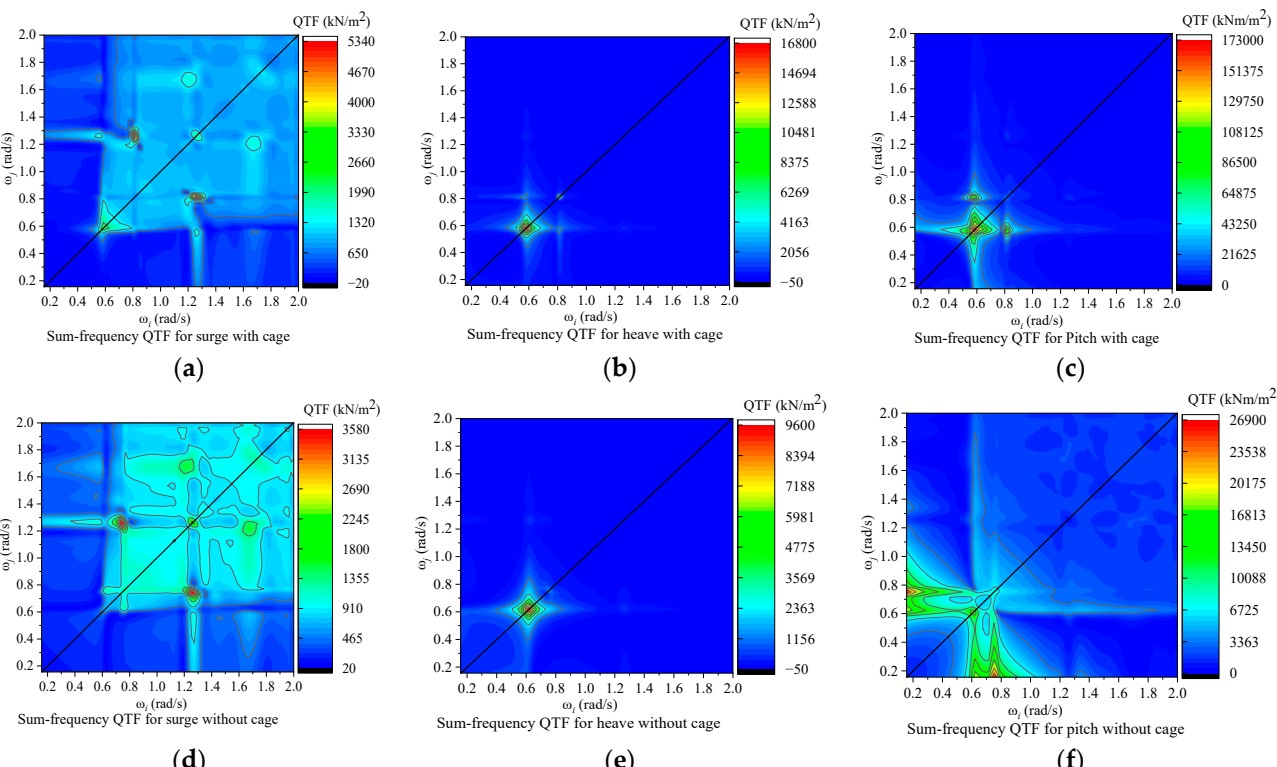

**Figure 12.** Second-order sum-frequency QTFs of the barge-type FOWT-AC and barge-type FOWT for a zero-degree wave heading: (**a**) Surge of FOWT-AC; (**b**) Heave of FOWT-AC; (**c**) Pitch of FOWT-AC; (**d**) Surge of FOWT; (**e**) Heave of FOWT; (**f**) Pitch of FOWT.

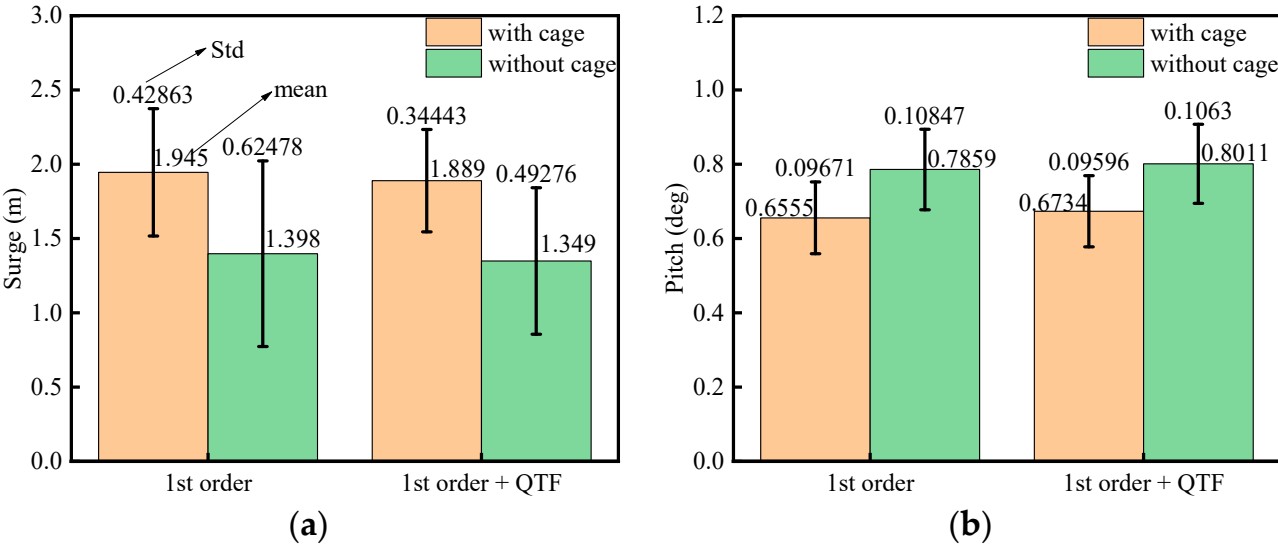

**Figure 13.** Comparison of motions for the barge-type FOWT-AC and barge-type FOWT at first order and first order with QTF conditions: (**a**) Mean and standard deviation of surge; (**b**) Mean and standard deviation of pitch.

### 4.2. Dynamic Analysis

For floating offshore structures, the various external loads and configuration difference are directly reflected in the motion response of the structure, thus the study of motion responses in six degrees of freedom is quite important. Herein, the environmental conditions with wind, wave and current in the same direction are chosen as the ideal environmental condition, which is the maximum external load case subjected by the structure. To explore the motion responses of the barge-type FOWT-AC and FOWT in the ideal environmental state, the time series results of 100 s after stable movement are outputted for EC1 as shown in Figure 14.

From Figure 14, it can be found that the presence of the aquaculture cage does have an effect on the motion responses of the structure. From Figure 14a, it is clear that the presence of the aquaculture cage makes the fluctuation amplitude of the surge motion smaller, but the mean value increases due to the Morison drag force on the nets. For roll, sway and yaw motions, Figure 14b,c,f show that the presence of the aquaculture cage reduces the fluctuation amplitude of the motions, but the mean value is still zero without change and the range of motion is small because the environmental loads are all inputted along the x-axis. As shown in Figure 14d, the fluctuation amplitude of the pitch motion is almost unchanged, but the overall motion angle becomes smaller due to the presence of the aquaculture cage. Thanks to the structure's eight suspension lines, there is sufficient recovery to allow for a small range of variation in pitch motion. Therefore, this results in the angle of pitch balance being less than 1° under the wind tilting moment. It can be seen in Figure 14e that the motion amplitude of the heave motion is almost unchanged when the aquaculture cage is present. It is also observed that the range of fluctuation in heave is smaller due to the large area of the platform baseboard compared to semi-submersible, spar, etc. In summary, the maximum value of the motion response of the barge-type FOWT is always larger than that of the barge-type FOWT AC except for the surge motion. Therefore, more attention should be paid to the surge motion in the design of the barge-type FOWT-AC to ensure that their leveling meets the specifications.

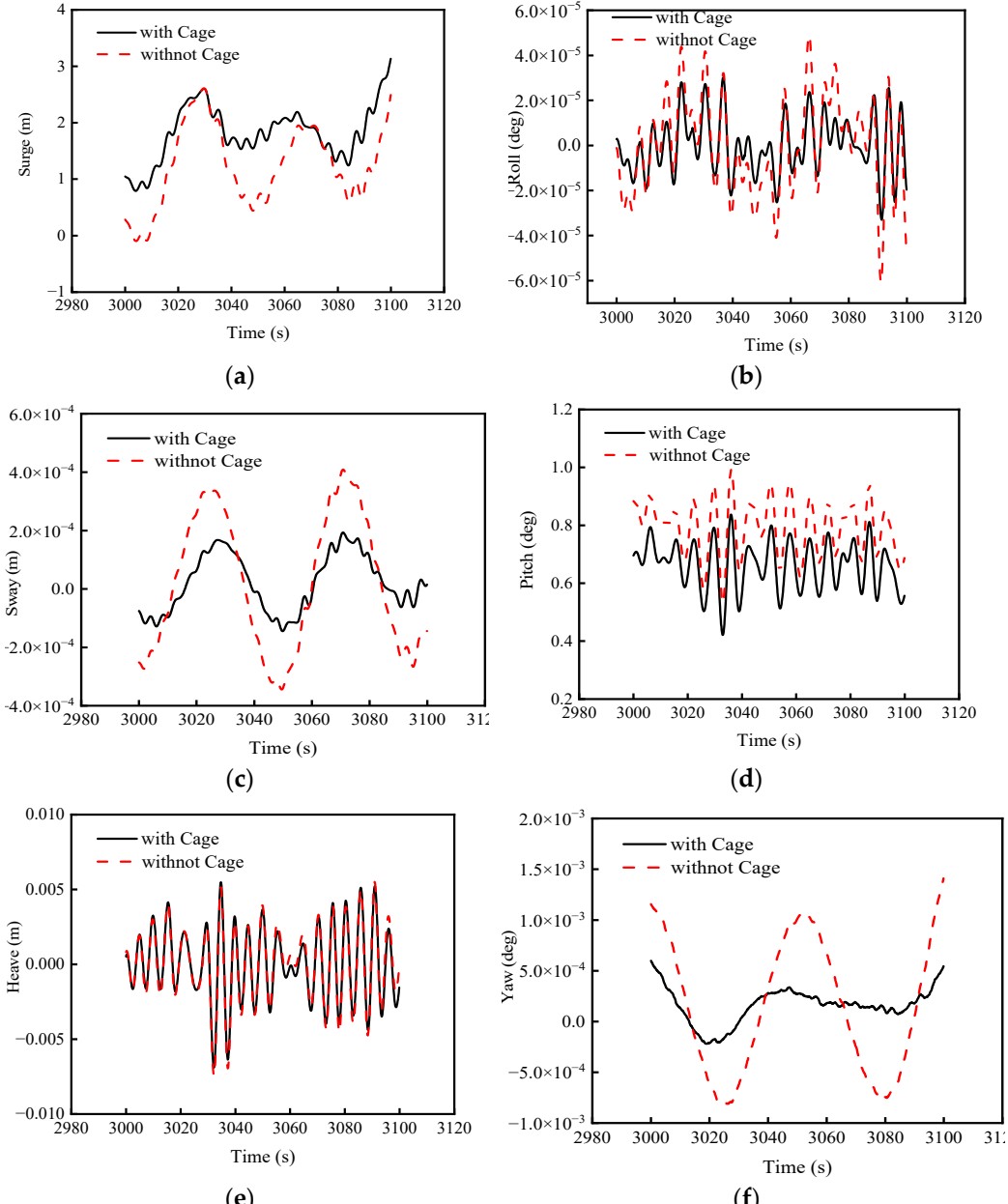

**Figure 14.** Comparison diagram of the motion response of the barge-type FOWT-AC and barge-type FOWT under EC1: (**a**) Time histories of surge motion; (**b**) Time histories of roll motion; (**c**) Time histories of sway motion; (**d**) Time histories of pitch motion; (**e**) Time histories of heave motion; (**f**) Time histories of yaw motion.

In order to better study the impact of the presence of aquaculture cages on the coupled structures, comparative analyses of surge and pitch motions of the floating structure with and without an aquaculture under varying wind, wave and current are carried out.

Comparisons of surge and pitch motions for the barge-type FOWT-AC and barge-type FOWT at different wind speeds are given in Figure 15. It can be seen from Figure 15a that the average value of the surge for barge-type FOWT-AC is larger than that of barge-type FOWT, but the standard deviation is opposite to the average value at different wind speeds. For Figure 15b, it can be observed that the average value and standard deviation of the pitch for barge-type FOWT is larger than that of barge-type FOWT-AC. This results trend is also consistent with Figure 14. The corresponding trend of surge and pitch motions for the same structure at different wind speeds is consistent with the variation of the wind speed-thrust

curve. Combining Figures 14 and 15, it can be found that the standard deviation of the six-degree-of-freedom motion responses of barge-type FOWT-AC at different wind speeds is smaller than that of the barge-type FOWT. As a result, the barge-type FOWT-AC is more stable in the operating operation. This is mainly because the presence of the aquaculture cage increases the hydrodynamic force on the coupled structure below the floating center.

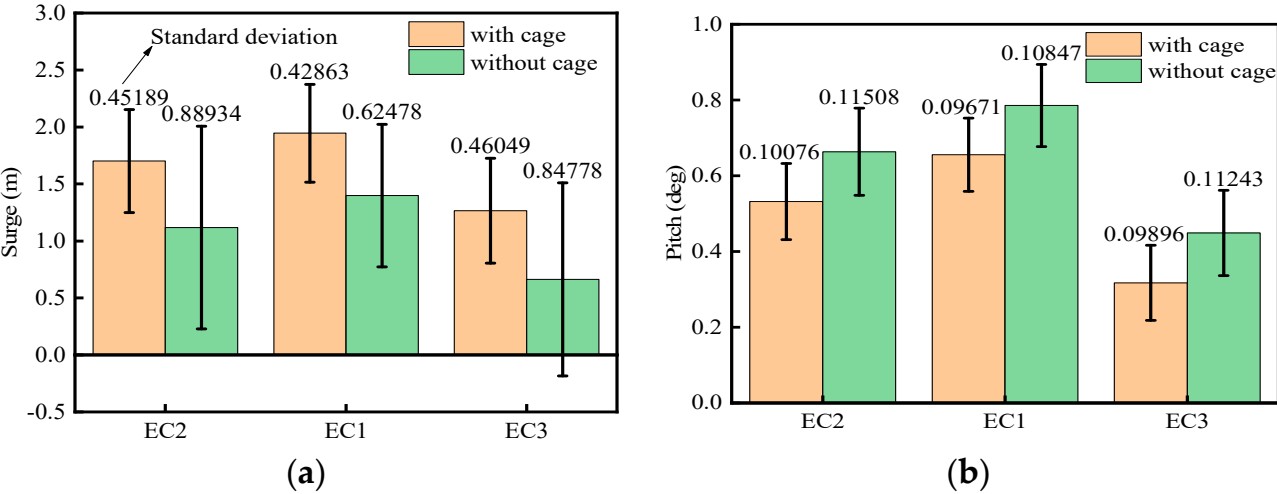

**Figure 15.** Comparison of motions for the barge-type FOWT-AC and barge-type FOWT at different wind speeds: (**a**) Mean and standard deviation of surge; (**b**) Mean and standard deviation of pitch.

Figure 16 shows a comparison of surge and pitch motions for the barge-type FOWT-AC and barge-type FOWT at different wave heights. From Figure 16a, it can be found that the mean and standard deviation values of the surge increase with the raising of wave height. From Figure 16b, it can be seen that the mean value of pitch decreases with the raising of the wave height, while the standard deviation values of pitch increase with the raising of wave height. The wave height of EC5 reaches 5.52 m, which already belongs to the extreme conditions. In this case, the mean value of surge for the barge-type FOWT-AC is 2.915 m, the standard deviation is 1.707 m, so the surge is less than 10% of the water depth, which is in line with the specifications the vast majority of the time. The mean value of pitch for the barge-type FOWT is 0.660°, the standard deviation is 4.812°, so the pitch is less than 10°, which is in line with the specification. Moreover, the comparison results in the means and standard deviations of surge and pitch for the structures with and without an aquaculture cage are the same as the conclusion obtained in the above for different wind speeds and wave conditions, indicating the stability of the designed floating structures.

Figure 17 illustrates that the presence of currents has a strong influence on the dynamic response of coupled structures, especially for the structure with an aquaculture cage. Figure 17a shows the offset of the equilibrium position of surge for the barge-type FOWT-AC with the current being 1 m/s is 3.54 times larger than that of no current, while it is 2.5 times larger for the case of the barge-type FOWT. From Figure 17b, it can be found that the mean value for the pitch of the barge-type FOWT-AC with the current being 1 m/s is 1.51 times larger than that of no current, while it is 1.73 times larger for the case of the barge-type FOWT. Therefore, the presence of currents significantly increases the mean value of the surge and pitch motions of the coupled structures. The presence of currents and the aquaculture cage decreases the standard deviation of the surge and pitch motion responses because the loads under the floating center of the structure in favor of the structural equilibrium are increased. The presence of currents has an effect on the amplitude of the motion and the equilibrium position of the floating structure [50], and these may be broadly associated with the potential flow and viscous effects [51].

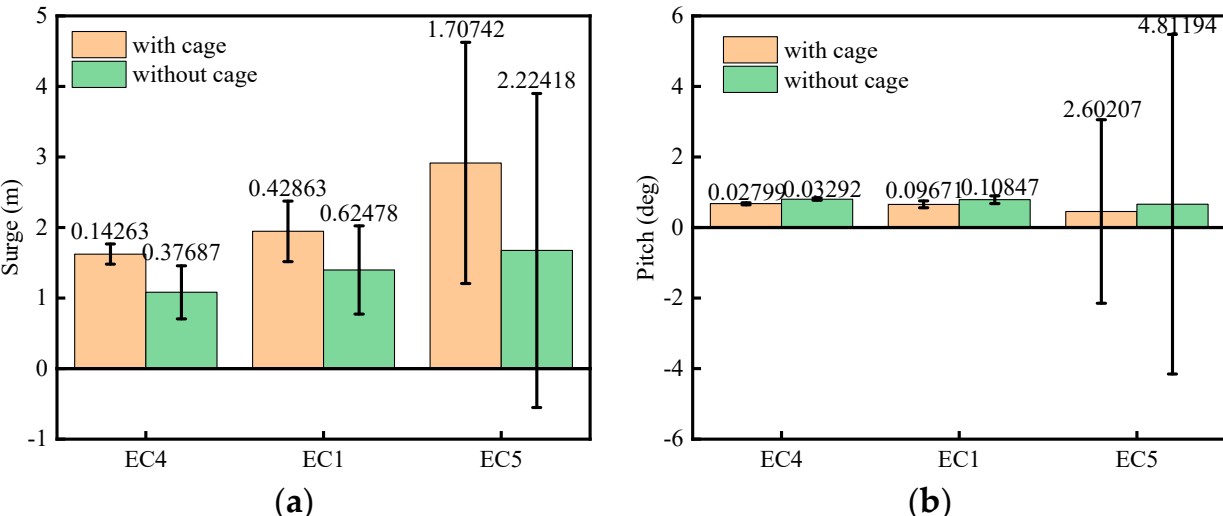

**Figure 16.** Comparison of motions for the barge-type FOWT-AC and barge-type FOWT at different wave height: (**a**) Mean and standard deviation of surge; (**b**) Mean and standard deviation of pitch.

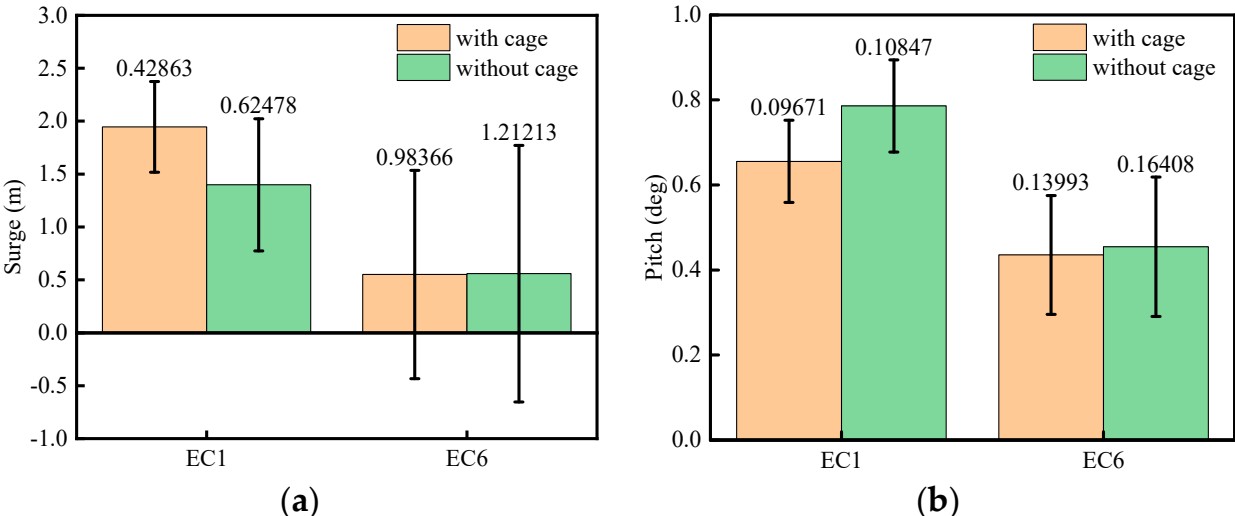

**Figure 17.** Comparison of motions for the barge-type FOWT-AC and barge-type FOWT in the presence or absence of currents: (**a**) Mean and standard deviation of surge; (**b**) Mean and standard deviation of pitch.

### 4.3. Dynamic Analysis of Coupled Structures with Mooring Line Failure

For the coupled structures proposed, the mooring lines can be divided into two groups, upstream and downstream, according to the environment conditions given in this study for the wind, wave and current with the same direction. The downstream suspension lines #1–#4 are in a looser state and therefore have less tension, while the upstream suspension lines #5–#8 are in a tighter state and therefore have more tension. Mooring line breakage is one of the most frequent failure conditions of FOWTs. To investigate the effect of mooring line breakage on the tension of other mooring lines and the motion response of the platform, the mooring lines located upstream (#8) and downstream (#3) are selected for the simulation of failure conditions in this paper.

As shown in Figure 18, breakage in one mooring line causes less tension on its diagonal mooring lines, more tension on the other mooring line at the same fairlead, and little change in tension on the remaining mooring lines. When mooring line #3 breaks, the mean values of tension of lines #1, #5 and #6 are smaller than that in the intact state, and the mean

values of tension of lines #2, #4, #7 and #8 are larger than that in the intact condition, where the change of line #4 is the largest. Under the condition of mooring line #3 breakage, the change of mean value of line #4 tension is within 20% compared with the normal operating condition, but the change of standard deviation is up to 127%. At this time, the mooring line #8 is used as the maximum tension of the remaining mooring lines, and the maximum amplitude of tension is 2297 kN for the barge-type FOWT-AC, which is smaller than the catalog breaking load of the mooring line 9319 kN. Hence, the rest of the mooring lines can still bear the force of the system normally with line #3 broken. When mooring line #8 breaks, the mean values of tension of lines #1–#4 are smaller than those in the intact state, and the mean values of tension are larger for lines #5–#7 than those in the intact state, where the change in tension for mooring line #7 is significantly larger by observing the change of mean value being within 60% and the standard deviation reaching up to 152%. At this time, mooring line #7 withstands the maximum tension of the whole mooring system, and the maximum amplitude of tension is 4316 kN, which is still smaller than the catalog breaking load of the mooring line 9319 kN. These variations increased a lot relative to line #3 breakage, because line #8 is subjected to more tension upstream before breaking.

Figure 19 shows the translations and rotational motions of the barge-type FOWT-AC under the mooring line #3 and line #8 breakage conditions. Mooring line breakage has a significant effect on the motion response of the coupled structure, especially when mooring line #8 breaks. The change of heave motion is not obvious because the result of heave motion is a change relative to the waterline, but the actual center of gravity has shifted upward after the mooring line breaking. When line #3 breaks, the offset of the equilibrium position of surge and pitch have decreased to a certain extent compared to the normal operating conditions. The motions of sway, roll and yaw have gone from almost zero to a certain magnitude due to the symmetry breaking under the condition of line #3 breakage. The offset of the equilibrium position of surge and pitch have increased to a certain extent compared to the intact state when the line #8 breaks. More notable sway, roll and yaw motions are caused by the mooring #8 breakage compared with the breakage of mooring line #3. Especially yaw, when mooring line #8 breaks, the motion magnitude value is enhanced more than 24 times compared with the breakage of mooring line #3.

Figure 20 presents the comparison of motion responses of the barge-type FOWT-AC and barge-type FOWT for different mooring line breakage. The standard deviation is smaller at the intact state for the two structures than for the failure state, and the standard deviation for the case of barge-type FOWT-AC is smaller than that of barge-type FOWT, implying that the barge-type FOWT-AC is more stable after mooring line breakage. The mean values of surge and pitch motions in the case of mooring line breakage are larger than those in the intact state for the two floating structures.

The presence of the aquaculture cage increases the relative variation of the surge and decreases the relative variation of the pitch when the mooring line #3 breaks, while the opposite change occurs when mooring line #8 breaks. In general, the most affected motion response of the coupled structure is the surge motion under the case of the mooring line #8 breakage, which has a mean value of 3.386 m and a standard deviation of 0.449 m.

The redundancy mooring system plays an important role in ensuring the safety of the entire coupled structure after a single mooring line breaks. When mooring line #3 breaks, its fairlead is restrained by mooring line #4 alone, so the tension of mooring line #4 will be increased. As shown in Figure 21, when the mooring line breaks, the tension of other mooring lines do not become larger or smaller immediately, but gradually changes according to the variable trend of the previous mooring line tension. This also shows that as long as the tension is less than the breaking strength, even if one mooring line breaks, the rest of the mooring lines will not be brittle fracture due to the sudden increase of tension [52,53].

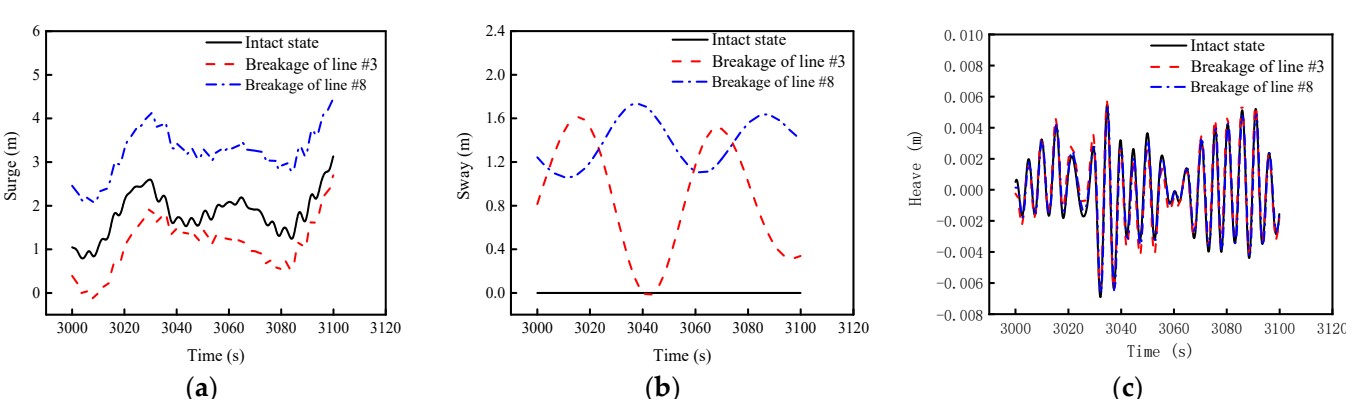

**Figure 18.** Mooring tension under different ECs for barge-type FOWT-AC: (**a**) Time histories of #1 tension; (**b**) Time histories of #2 tension; (**c**) Time histories of #3 tension; (**d**) Time histories of #4 tension; (**e**) Time histories of #5 tension, (**f**) Time histories of #6 tension; (**g**) Time histories of #7 tension; (**h**) Time histories of #8 tension.

**Figure 19.** *Cont.*

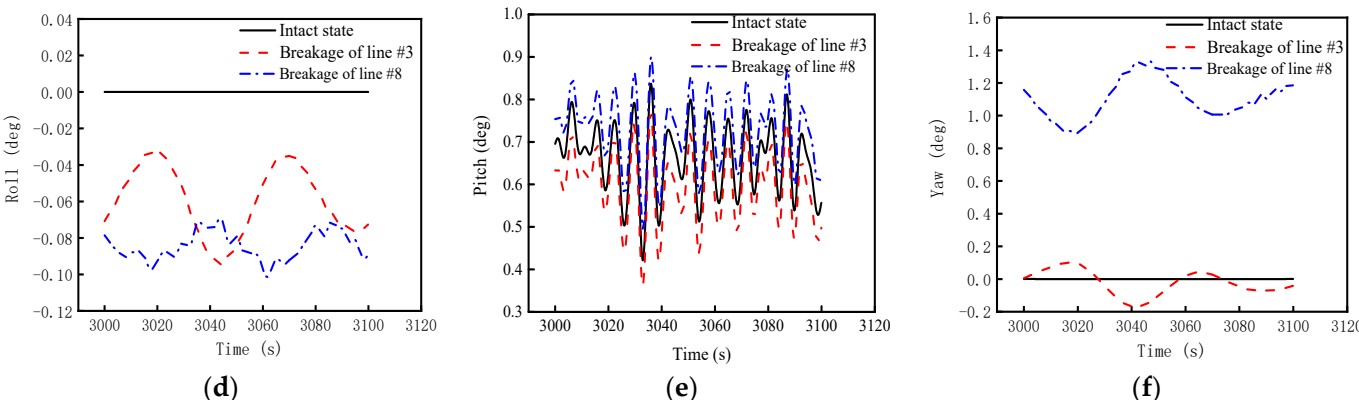

**Figure 19.** Motion response under different ECs for barge-type FOWT-AC: (**a**) Time histories of surge; (**b**) Time histories of sway; (**c**) Time histories of heave; (**d**) Time histories of roll; (**e**) Time histories of pitch; (**f**) Time histories of yaw.

**Figure 20.** Comparison of motions for the barge-type FOWT-AC and barge-type FOWT under different ECs: (**a**) Surge at breakage of #3; (**b**) Pitch at breakage of #3; (**c**) Surge at breakage of #8; (**d**) Pitch at breakage of #8.

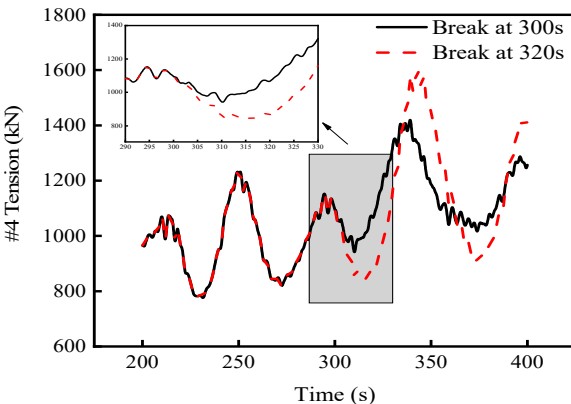

**Figure 21.** Comparison of #4 tension when #3 breaks at different moments.

### 5. Conclusions

This paper focuses on proposing a new barge-type FOWT-AC structure and comparing the numerical simulation results with the barge-type FOWT considering first- and second-order wave loads, wind loads and current loads of the environmental conditions in the South China Sea at normal operating conditions and fault conditions. The effects of the aquaculture cage on the dynamic responses of the barge-type FOWT are explored. From this study, the following conclusions are drawn:

(1) The presence of the aquaculture cage has improved the natural period of the heave, pitch, and roll. Except for surge, the maximum values of the motions of the barge-type FOWT-AC are smaller than barge-type FOWT. There is a reduction in the standard deviation of the motion response due to the presence of the aquaculture cage. Therefore, the barge-type FOWT-AC is generally more stable during normal operation.

(2) The effect of second-order wave excitation on the dynamic responses of floating structures is considered in detail, and it is noted that the differential-frequency relative to the sum-frequency has a larger range of effects on the coupled structure. Because the second-order wave loads are not of the same order of magnitude as the first-order wave loads, they have little effect on the motion response of the coupled structures when the structural resonance is not induced.

(3) By comparing the influence of the presence or absence of currents, it can be concluded that the presence of currents has a strong influence on the dynamic response of coupled structures, making the motion response increase significantly because of the increasing of the external loads on the barge-type FOWT-AC. The surge motion of the barge-type FOWT-AC can even reach up to 3.54 times greater when the current is 1 m/s than when there is no current, so the effect of currents cannot be neglected in the hydrodynamic analysis.

(4) The breakage of one mooring line causes less tension on its diagonal mooring lines, more tension on the other mooring line at the same fairlead, and little change in tension on the remaining mooring lines. When the upstream mooring line breaks, it has a greater effect on the motion response of the coupled structure and the mooring tension than the accident of downstream mooring line breakage. More importantly, compared to barge-type FOWT, the standard deviation of the barge-type FOWT-AC is smaller and relatively more stable after mooring line breakage.

### 6. Future Work

In this paper, some hydrodynamic characteristics and motion responses of the FOWT-AC structure are initially investigated using numerical simulations, but there are still many inadequacies to be investigated in the future.

First of all, the occurrence of extreme environmental conditions has increased due to climate change. The probability of breaking waves occurring that are induced by extreme conditions will certainly increase. The external loads on marine structures under the action

of breaking waves and unbroken waves are different by considering the air entrapment effect [54] and the significant impulsive/slamming loads [55] exerted by breaking waves. Therefore, it is important to investigate the changes in the motion response of marine structures subjected to breaking waves. However, predicting the air entrapment effect [54] and high-frequency impulsive/slamming loads [55] accurately in dealing with the breaking wave scenario is a challenge, and whether the currently used simulation methods are still available will be considered in future works.

Secondly, the wave-in-platform loads are a very complex research field due to the large horizontal and uplift forces observed in past studies [56–61], the nonlinear interaction of the wave with the air below the deck [59,60], and the application of a large overturning moment [55,61] that could lead to a concentration of forces in the offshore supports. None of these complex loads were addressed in this research study, but they will be considered in future research works.

Finally, in the current study, the wave direction is normal for the offshore surface of the barge platform, which is an idealized case where the FOWT-AC structure can withstand the maximum external load. However, the actual wave and wind loads may propagate from any angle; thus, load input conditions for different angles will also be investigated in future works.

**Author Contributions:** Conceptualization, Y.Z., H.Z., X.L. and W.S.; methodology, Y.Z.; software, Y.Z.; validation, Y.Z. and H.Z.; formal analysis, Y.Z.; investigation, Y.Z. and H.Z.; data curation, Y.Z.; writing—original draft preparation, Y.Z.; writing—review and editing, H.Z. and X.L.; supervision, H.Z., X.L. and W.S.; project administration, W.S.; funding acquisition, X.L. All authors have read and agreed to the published version of the manuscript.

**Funding:** This research work was financially supported by the National Natural Science Foundation of China, Grant No. is 51939002.

**Institutional Review Board Statement:** Not applicable.

**Informed Consent Statement:** Not applicable.

**Data Availability Statement:** Not applicable.

**Conflicts of Interest:** The authors declare no conflict of interest.

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
