# Peer review of "Hydrodynamic Responses of a Barge-Type Floating Offshore Wind Turbine Integrated with an Aquaculture Cage"

_jmse, doi:10.3390/jmse10070854_

Round 1
Reviewer 1 Report
Please see attached file.

Reviewer 2 Report
Good and valuable paper, ready for publication after some minor changes that authors have to accomplish.
· First of all a concise list of the objectives of the paper should be added at the end of the introductory section so that the reader can more easily identify the composition of the paper;
· Regarding the equations, a slightly larger font should be used for indices, as well as adjusting the space after the equation (there are large spaces between the equation and the next row);
· The figures may need to be placed in the center for a better look of the page, but please also consult the author's formatting rules for the journal;
· Some extra-lines added between tables (lines 251, 253, 255);
· Table 5 seems to exceed the page limit. Please re-dimension the table;
· The same situation for figures 10-20. You can adjust the spaces within the figures table;
· Some text font differences are met at conclusion section (line 511);
· The title and subtitle of the Results and Discussion section (page 12) should start on a new page;
· Text references to figures and tables are written with or without the bold option. They need to be standardized.

Round 2
Reviewer 1 Report
The authors have addressed properly the majority of the reviewer's comments by providing additional explanations, clarifications and discussion of the limitations. The new changes have improved the value of the manuscript and is therefore suggested for publication.
The only minor comment is that the phrase "research center" does not seem to be correct and should be replaced by a more appropriate phrase, such as, "research study".
Author Response
Detailed reply to reviewer’s comments for reviewer:
Comment: The only minor comment is that the phrase "research center" does not seem to be correct and should be replaced by a more appropriate phrase, such as, "research study".
Response to comment: Thanks to the reviewer for pointing out the problem, we have made the following changes: “None of these complex loads were addressed in this research study, but will be considered gradually in future research works.” at L595-596, and the correction is marked in red in the manuscript.